# Practical Computational Power of Linear Transformers and Their Recurrent and Self-Referential Extensions

**Kazuki Irie**[1][†]  **Róbert Csordás**[2]  **Jürgen Schmidhuber**[2,3]

[1]Harvard University  [2]The Swiss AI Lab IDSIA, USI & SUPSI  [3]AI Initiative, KAUST

kirie@fas.harvard.edu  {robert,juergen}@idsia.ch

## Abstract

Recent studies of the computational power of recurrent neural networks (RNNs) reveal a hierarchy of RNN architectures, given real-time and finite-precision assumptions. Here we study auto-regressive Transformers with linearised attention, a.k.a. linear Transformers (LTs) or Fast Weight Programmers (FWPs). LTs are special in the sense that they are equivalent to RNN-like sequence processors with a fixed-size state, while they can also be expressed as the now-popular self-attention networks. We show that many well-known results for the standard Transformer directly transfer to LTs/FWPs. Our formal language recognition experiments demonstrate how recently proposed FWP extensions such as recurrent FWPs and self-referential weight matrices successfully overcome certain limitations of the LT, e.g., allowing for generalisation on the parity problem. Our code is public.[1]

## 1  Introduction

Recurrent neural networks (RNNs) are Turing-complete, given assumptions of infinite precision and unbounded computation (Siegelmann and Sontag (1991) for the sigmoid activation; Chen et al. (2018) for ReLU; see also an alternative setting of Chung and Siegelmann (2021)). This insight has a certain theoretical value, however, its implication for practical, real-world scenarios is limited. In contrast, many recent works (e.g., Weiss et al. (2018b); Suzgun et al. (2019a); Merrill et al. (2020); Bhattamishra et al. (2020a)) strive to obtain results of practical relevance on the computational power of sequence-processing neural networks (NNs). Typical assumptions of such studies are finite precision and "input-bound" computation (i.e., the number of computational steps is determined by and limited to the number of input tokens; a.k.a. real-time assumption), essentially reflecting the reality of practical

RNNs. Although such analyses typically further assume models to be *saturated* (Merrill et al. (2020); which is not always the case for real-world RNNs), they provide useful results and insights that can be empirically confirmed, e.g., on formal language recognition tasks. For example, the long short-term memory (LSTM; Hochreiter and Schmidhuber (1997)) can learn to solve and generalise on certain counter-language tasks (Gers and Schmidhuber (2001); Schmidhuber et al. (2002); Weiss et al. (2018b); Suzgun et al. (2019a)), unlike its simplified versions such as the gated recurrent unit (GRU; Cho et al. (2014); Gers et al. (2000)), the Quasi-RNN (Bradbury et al. (2017); see Merrill et al. (2020)) or the simple sigmoid RNN (Elman, 1988). Tools from the theory of computation (see, e.g., Sipser (1996)) provide a formalism to compare the practical computational power of these NN architectures, allowing for categorisation into a hierarchy (Merrill et al. (2020); see Appendix A for further references).

More recent works (Bhattamishra et al., 2020a,b; Ebrahimi et al., 2020; Yao et al., 2021) derive results for the now-popular Transformer (Vaswani et al., 2017). For example, Bhattamishra et al. (2020a) show (we review in Sec. 3) that Transformers can learn to solve and generalise on certain counter-language tasks (including the context-sensitive $a^n b^n c^n$), while they fail to learn certain simple regular languages (many examples of which are *non-star-free* languages) including the parity task (see also Chiang and Cholak (2022)). For further theoretical studies, we refer to, e.g., Hahn (2020); Merrill et al. (2022); Hao et al. (2022).

Here we focus on auto-regressive Transformers with *linearised attention* a.k.a. linear Transformers (LTs; Katharopoulos et al. (2020); Choromanski et al. (2021); Peng et al. (2021)) or Fast Weight Programmers (FWPs; Schmidhuber (1991, 1992a); Schlag et al. (2021); Irie et al. (2021)). LTs are special, as they can be equivalently expressed as

---

[†] Work done at IDSIA.

[1] https://github.com/IDSIA/fwp-formal-lang

an RNN-like sequence processor with a constant-size state (the FWPs from the 1990s; Schmidhuber (1991, 1993)), while they are originally formulated as self-attention networks. This property is interesting for discussions of computational power, since it removes one of the classic distinctions between Transformers and RNNs: RNNs are "automata-like" constant-size stateful models, while Transformers are not. Building upon Weiss et al. (2018b); Bhattamishra et al. (2020a); Merrill et al. (2020), we show that many existing results on Transformers directly transfer to LTs/FWPs. Also, prior work proposes several extensions to LTs inspired by its RNN-like form, including recurrence (Irie et al., 2021) and self-reference (Irie et al., 2022c), showing their practical benefits on actual real-world tasks (e.g., reinforcement learning in game environments). Here we demonstrate how both recurrent and self-referential extensions enhance LTs' practical computational power on formal language recognition tasks.

## 2 Background

Here we briefly review LTs and FWPs. Let $d_{\text{in}}$, $d_{\text{out}}$, $d_{\text{key}}$, $t$ be positive integers, and $\otimes$ denote outer product. An FWP (Schmidhuber, 1991, 1992a) is a sequence-processing NN that, at each time step $t$, transforms an input $\boldsymbol{x}_t \in \mathbb{R}^{d_{\text{in}}}$ to an output $\boldsymbol{y}_t \in \mathbb{R}^{d_{\text{out}}}$ as follows:

$$\boldsymbol{q}_t, \boldsymbol{k}_t, \boldsymbol{v}_t = \boldsymbol{W}^{\text{slow}} \boldsymbol{x}_t \qquad (1)$$

$$\boldsymbol{W}_t = \boldsymbol{W}_{t-1} + \boldsymbol{v}_t \otimes \phi(\boldsymbol{k}_t) \qquad (2)$$

$$\boldsymbol{y}_t = \boldsymbol{W}_t \phi(\boldsymbol{q}_t) \qquad (3)$$

where $\boldsymbol{k}_t, \boldsymbol{q}_t \in \mathbb{R}^{d_{\text{key}}}$, $\boldsymbol{v}_t \in \mathbb{R}^{d_{\text{out}}}$, and $\boldsymbol{W}^{\text{slow}} \in \mathbb{R}^{(2d_{\text{key}}+d_{\text{out}}) \times d_{\text{in}}}$ is a weight matrix (the "slow" weights), and $\phi$ is an activation function. The "fast" weight matrix $\boldsymbol{W}_t \in \mathbb{R}^{d_{\text{out}} \times d_{\text{key}}}$ is initially set to 0, i.e., $\boldsymbol{W}_0 = 0$. This can be viewed as a system of two NNs where one net (the slow net; Eq. 1) learns to "program" the fast net (Eq. 3) by generating its weight changes (Eq. 2). This $\boldsymbol{x}_t$-to-$\boldsymbol{y}_t$ transformation can be also expressed as *linear attention* using $\phi(\boldsymbol{k}_t), \boldsymbol{v}_t, \phi(\boldsymbol{q}_t)$ as key, value, query vectors (Katharopoulos et al. (2020); Schlag et al. (2021); Ba et al. (2016); see also our brief review in Appendix B). To be more specific, such FWPs correspond to *unnormalised* LTs. LTs with *normalised* linear attention (NLTs; Katharopoulos et al. (2020); Choromanski et al. (2021); Peng et al. (2021)) use

the following additional computation:

$$\boldsymbol{z}_t = \boldsymbol{z}_{t-1} + \phi(\boldsymbol{k}_t) \qquad (4)$$

$$\boldsymbol{y}_t = \frac{1}{\boldsymbol{z}_t \cdot \phi(\boldsymbol{q}_t)} \boldsymbol{W}_t \phi(\boldsymbol{q}_t) \qquad (5)$$

where $\boldsymbol{z}_t \in \mathbb{R}^{d_{\text{key}}}$ with $\boldsymbol{z}_0 = 0$, and $\cdot$ denotes dot product, i.e., $\boldsymbol{z}_t \cdot \phi(\boldsymbol{q}_t) \in \mathbb{R}$, replacing Eq. 3. In practice, this normalisation can be removed without loss of performance (Schlag et al., 2021; Irie et al., 2021), which is convenient as no extra vector $\boldsymbol{z}_t$ needs to be stored. All standard Transformer components including feedforward blocks, residual connections, and layer-norm are used in LTs.

This equivalence has inspired a series of extensions to Transformers. Here we highlight three such examples: delta-rule, recurrence, and self-reference, which we study in Sec. 4.

**Delta-rule.** Schlag et al. (2021) replace the purely additive update rule of Eq. 2 by the classic delta-rule for error correction (Widrow and Hoff, 1960); Eq. 8 below. The slow weight matrix in the resulting model, called DeltaNet, is $\boldsymbol{W}_{\text{slow}} \in \mathbb{R}^{(2*d_{\text{key}}+d_{\text{out}}+1) \times d_{\text{in}}}$ that also generates a dynamic learning rate $\beta_t \in \mathbb{R}$ (to which we apply the sigmoid function $\sigma$). With the delta rule, $\phi$'s output elements need to be positive and sum up to one (e.g., we use softmax) for stability.

$$\boldsymbol{q}_t, \boldsymbol{k}_t, \boldsymbol{v}_t, \beta_t = \boldsymbol{W}^{\text{slow}} \boldsymbol{x}_t \qquad (6)$$

$$\bar{\boldsymbol{v}}_t = \boldsymbol{W}_{t-1} \phi(\boldsymbol{k}_t) \qquad (7)$$

$$\boldsymbol{W}_t = \boldsymbol{W}_{t-1} + \sigma(\beta_t)(\boldsymbol{v}_t - \bar{\boldsymbol{v}}_t) \otimes \phi(\boldsymbol{k}_t) \qquad (8)$$

Note that this introduces an extra dependency to LTs; the update term $\sigma(\beta_t)(\boldsymbol{v}_t - \bar{\boldsymbol{v}}_t) \otimes \phi(\boldsymbol{k}_t)$ in Eq. 8 is a function of $\boldsymbol{W}_{t-1}$ unlike in Eq. 2. We'll empirically illustrate how this modification introduces an explicit forget mechanism to the LT, using the "reset Dyck-1" language (Sec. 4).

**Recurrence.** One trivial extension to the LTs/DeltaNet above is to add "proper recurrent connections" by feeding back the output $\boldsymbol{y}_{t-1}$ from the previous step $t-1$ as an input at step $t$ (for other recurrent extensions we refer to Irie et al. (2021)). The resulting Recurrent DeltaNet (Irie et al., 2021) (or Recurrent Delta) is obtained by replacing Eq. 6 in the DeltaNet by:

$$\boldsymbol{q}_t, \boldsymbol{k}_t, \boldsymbol{v}_t, \beta_t = \boldsymbol{W}^{\text{slow}}[\boldsymbol{x}_t, \tanh(\boldsymbol{y}_{t-1})] \qquad (9)$$

where $\boldsymbol{W}_{\text{slow}} \in \mathbb{R}^{(2*d_{\text{key}}+d_{\text{out}}+1) \times (d_{\text{in}}+d_{\text{out}})}$, and $[\boldsymbol{x}_t, \tanh(\boldsymbol{y}_{t-1})] \in \mathbb{R}^{d_{\text{in}}+d_{\text{out}}}$ denotes concatenation of the two vectors.

**Self-Reference.** Another extension of the DeltaNet above is the *modern* self-referential weight matrix (SRWM). Motivated by recursive self-improvement (Good, 1965; Schmidhuber, 1987) and the original SRWM (Schmidhuber, 1992b), Irie et al. (2022c) extend the FWP that generates weight changes for another NN to obtain an NN that modifies itself. At each time step $t$, an SRWM $\boldsymbol{W}_{t-1} \in \mathbb{R}^{(d_{\text{out}}+2*d_{\text{in}}+1) \times d_{\text{in}}}$ transforms an input $\boldsymbol{x}_t \in \mathbb{R}^{d_{\text{in}}}$ to an output $\boldsymbol{y}_t \in \mathbb{R}^{d_{\text{out}}}$, and updates itself to $\boldsymbol{W}_t$ as follows:

$$\boldsymbol{y}_t, \boldsymbol{k}_t, \boldsymbol{q}_t, \beta_t = \boldsymbol{W}_{t-1}\boldsymbol{x}_t \tag{10}$$

$$\boldsymbol{v}_t = \boldsymbol{W}_{t-1}\phi(\boldsymbol{q}_t); \; \bar{\boldsymbol{v}}_t = \boldsymbol{W}_{t-1}\phi(\boldsymbol{k}_t) \tag{11}$$

$$\boldsymbol{W}_t = \boldsymbol{W}_{t-1} + \sigma(\beta_t)(\boldsymbol{v}_t - \bar{\boldsymbol{v}}_t) \otimes \phi(\boldsymbol{k}_t) \tag{12}$$

where $\boldsymbol{v}_t, \bar{\boldsymbol{v}}_t \in \mathbb{R}^{(d_{\text{out}}+2*d_{\text{in}}+1)}$, $\boldsymbol{q}_t, \boldsymbol{k}_t \in \mathbb{R}^{d_{\text{in}}}$ and $\beta_t \in \mathbb{R}$. The initial values $\boldsymbol{W}_0$ are the only trainable parameters of this layer. Note that it is straightforward to further extend this model with recurrence as in Eq. 9. Here we focus on this specific self-referential extension.

## 3 Expressive Power of LTs

Here we revisit several existing results on the practical computational power of Transformers for normalised LTs (NLTs; Eqs. 1-2;4-5). Results in this section directly build upon prior work (Bhattamishra et al., 2020a; Merrill et al., 2020). As we'll see, some of the results are not obvious from Eqs. 1-2;4-5. However, their connection to Transformers allows us to trivially derive them. While one can come up with certain custom positional encoding methods that empower Transformers to specifically recognise certain languages, here we focus on generic Transformers without positional encoding (Bhattamishra et al., 2020a; Irie et al., 2019; Tsai et al., 2019).

We start by noticing that the hidden "state" update of NLTs (Eq. 2) is element-wise. This is reminiscent of simplified LSTMs such as Quasi-RNNs (Bradbury et al., 2017), which are known to be limited (Merrill et al., 2020): in particular, Quasi-RNNs are rationally recurrent (Peng et al. (2018)). However, we have the following result:

**Proposition 3.1** ("Rational Recurrence"). *NLTs are not rationally recurrent.*

*Proof.* Merrill et al. (2020)'s proof by construction for their Theorem 15 remains valid for NLTs. □

It should be noted that the actual output of NLTs is $\boldsymbol{y}_t$ (Eq. 5), not $\boldsymbol{W}_t$. In fact, NLTs can recognise certain counter languages, inheriting the properties of the Transformer:

**Proposition 3.2** ("Simple Counter Languages"). *NLTs can recognise certain counter languages.*

*Proof.* Bhattamishra et al. (2020a)'s proof for their Proposition 4.1 is valid for NLTs: NLTs can recognise Shuffle-Dyck languages. □

However, similar to Transformers, NLTs are fundamentally limited:

**Proposition 3.3** ("Regular Languages"). *NLTs can not recognise certain regular languages.*

*Proof.* Bhattamishra et al. (2020a)'s proof for their Lemma C.4 remains valid for NLTs; NLTs can not recognise the regular language $(aa)^*$. □

Finally, we comment on the "state complexity" as defined by Merrill et al. (2020):

**Proposition 3.4** ("State complexity"). *The state complexity of a single-layer NLT is $O(\log(n))$ (same as the regular self-attention and LSTM).*

*Proof.* Merrill et al. (2020)'s proof for Theorem 16 remains valid for normalised linear attention. □

Given the original proofs by Bhattamishra et al. (2020a) and Merrill et al. (2020), the proofs above are straightforward for *normalised* LTs. For further discussions on these proofs and their extension for *unnormalised* LTs (Eqs. 1-3), we refer to Appendix C.1. In sum, these statements on the expressiveness of Transformers remain valid for both normalised and unnormalised LTs.

## 4 Experiments

Here we provide several empirical results on capabilities and limits of *unnormalised* LTs/FWPs and their extensions, using formal languages.

### 4.1 Tasks

We evaluate LT models on formal language recognition tasks using several non-star-free regular languages—parity, $(aa)^*$, $(abab)^*$—, and counter languages—$a^n b^n$, $a^n b^n c^n$, Shuffle-2, and reset Dyck-1. This choice is guided by Bhattamishra et al. (2020a)'s results on the standard Transformers to specifically evaluate LTs' capabilities and limits. Following prior work (Gers and Schmidhuber, 2001), for $a^n b^n$ and $a^n b^n c^n$,

Table 1: Accuracies of various models on the formal language recognition tasks.

| | Non-Star-Free Regular | | | | | | Counter | | | | | |
| | Parity | | $(aa)^*$ | | $(abab)^*$ | | $a^n b^n$ | | $a^n b^n c^n$ | | Shuffle-2 | |
| Model | Bin0 | Bin1 | Bin0 | Bin1 | Bin0 | Bin1 | Bin0 | Bin1 | Bin0 | Bin1 | Bin0 | Bin1 |
|---|---|---|---|---|---|---|---|---|---|---|---|---|
| LSTM | **100.0** | **100.0** | **100.0** | **100.0** | **100.0** | **100.0** | **100.0** | **100.0** | **100.0** | **100.0** | **100.0** | **100.0** |
| e-LSTM | **100.0** | **100.0** | **100.0** | **100.0** | **100.0** | **100.0** | **100.0** | 90.0 | **100.0** | 22.0 | **100.0** | 85.7 |
| Transformer | 47.1 | 0.1 | 0.0 | 0.0 | 0.0 | 0.0 | **100.0** | **100.0** | **100.0** | **100.0** | **100.0** | **100.0** |
| Linear | 77.9 | 0.2 | 0.0 | 0.0 | 0.0 | 0.0 | **100.0** | **100.0** | **100.0** | **100.0** | **100.0** | **100.0** |
| DeltaNet | 97.3 | 11.8 | 0.0 | 0.0 | 0.0 | 0.0 | **100.0** | **100.0** | **100.0** | **100.0** | **100.0** | **100.0** |
| Recurrent Delta | **100.0** | **100.0** | **100.0** | **100.0** | **100.0** | **100.0** | **100.0** | **100.0** | **100.0** | **100.0** | **100.0** | **100.0** |
| SRWM | **100.0** | **100.0** | **100.0** | **100.0** | **100.0** | **100.0** | **100.0** | **100.0** | **100.0** | **100.0** | **100.0** | **100.0** |

we define the language recognition task as the next character prediction task. For example, for the context-free language $a^n b^n$, if the input to the model is aaabbb, the model has to output NNNbbS where N denotes "*cannot-predict-yet*" token, and S denotes the sequence-end token. Appendix C.2 contains corresponding descriptions for other tasks. We train models on positive examples of up to a certain length, and validate them on positive examples with longer lengths. We denote the corresponding data splits as "Bin0" (sequences with lengths seen during training) and "Bin1" (longer sequences).

For evaluation, for each position of the sequence, we define the model prediction as the token that is the most likely according to the model. We report accuracy on the sequence level; we count a sequence as correctly recognised only if the model prediction is correct for all tokens in the sequence. Further experimental details (e.g., hyperparameters) can be found in Appendix C.2.

## 4.2 Results

**Main Results.** Table 1 shows our main results. The top part of the table shows three reference baselines: LSTM, Transformer, and an LSTM with element-wise recurrence and tied input-forget gate (denoted as e-LSTM; see details in Appendix C.3). Table 1/Left shows the results for the regular language tasks: parity, $(aa)^*$, and $(abab)^*$, on which Bhattamishra et al. (2020a) report Transformers to fail. Inheriting their properties, the standard LT fail on all these tasks (recall, however, as stated by Bhattamishra et al. (2020a): non-star-free regular languages are not the strict set of regular languages on which Transformers fail). We also confirm that the delta rule is not enough to help LTs succeed in these tasks. In contrast, both the recurrent (Recurrent Delta) and self-referential (SRWM) extensions successfully solve and generalise on these tasks.

Table 2: Accuracies of single-layer models.

| | Dyck-1 | | Reset Dyck-1 | |
| Model | Bin0 | Bin1 | Bin0 | Bin1 |
|---|---|---|---|---|
| Linear | **100.0** | **100.0** | 44.5 | 41.1 |
| DeltaNet | **100.0** | **100.0** | **100.0** | **100.0** |
| Recurrent Delta | **100.0** | **100.0** | **100.0** | **100.0** |
| SRWM | **100.0** | **100.0** | **100.0** | **100.0** |

We also confirm that all LT variants can learn representative counter languages that the original Transformer can learn (Table 1/Right). One interesting empirical trend (not reflected in Table 1) is that the base Transformer and LT tend to more easily find solutions that generalise. For DeltaNet, Recurrent Delta and SRWM, many configurations achieve 100% accuracy on Bin0, without achieving exactly 100% on Bin1.

**Reset Dyck-1.** The main experiment above does not emphasize the benefits of the delta-rule which by itself cannot help LTs to recognise parity, $(aa)^*$ and $(abab)^*$. However, the delta rule is typically reported to be crucial across many practical tasks (including reinforcement learning in game environments (Irie et al., 2021), image generation (Irie and Schmidhuber, 2023), or long time-series classification (Irie et al., 2022b)). Here we use reset Dyck-1 to illustrate its benefits. Bhattamishra et al. (2020a) prove that a one-layer self-attention network cannot recognise reset Dyck-1 as it has no mechanism to rewrite memory. As shown in Table 2, the delta-rule by itself allows LTs to recognise this language.

## 5 Outlook: Self-Modifying Automata

Here we discuss a potentially interesting perspective for further studying SRWM models. Self-Modifying Automata (SMAs) and Self-Modifying

Finite Automata (SMFAs; Rubinstein and Shutt (1993, 1995); Shutt (1995); Wang et al. (1999))[2] are "FAs" with capabilities to modify their transition function at runtime. Despite its name containing "finite automata," they are provably computationally more powerful than FAs: the least restricted versions thereof are Turing-complete, certain restricted ones can still recognise certain context-sensitive languages (meta-linear languages). It may be interesting to connect SMFAs with SR-WMs in future work. For example, we may try to extract SMFAs from SRWMs trained on certain meta-linear/counter languages.

## 6 Conclusion

We discuss the computational power of Transformers with linearised attention in the context of formal languages. We show that such linear Transformers (LTs) a.k.a. Fast Weight Programmers (FWPs) inherit several capabilities and limitations of Transformers. We demonstrate that their recurrent and self-referential extensions successfully overcome some of those limitations. We hope this work will inspire the development of formally more powerful Transformer models.

## Limitations

Our study follows prior work on the computational power of RNNs, in particular, Bhattamishra et al. (2020a)'s results on Transformers, and Merrill et al. (2020)'s discussion of state-complexity and rational recurrence. Naturally, this is not an exhaustive study of LT properties, and we cannot definitively compare the expressivity of LTs to the one of standard Transformers solely based on what is presented here. Also, while it is rather obvious that the recurrent extension enhances the computational power of LTs, future work should provide more insights on the power of self-reference; see also our outlook Sec. 5 on *self-modifying automata*. A comparison of such models to memory-augmented RNNs (Graves et al., 2016; Suzgun et al., 2019b; Delétang et al., 2023) is also left for future work.

We focus on eight tasks by Bhattamishra et al. (2020a) to illustrate the capabilities and limitations of Transformers (and thus, those of LTs). Future work will extend experimental results to more

diverse tasks, e.g., those presented in Delétang et al. (2023).

## Acknowledgements

This research was partially funded by ERC Advanced grant no: 742870, project AlgoRNN, and by Swiss National Science Foundation grant no: 200021_192356, project NEUSYM. We are thankful for hardware donations from NVIDIA and IBM. The resources used for this work were partially provided by Swiss National Supercomputing Centre (CSCS) project d123.

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

# A  RNNs and Theory of Computation

RNNs have been related to Finite Automata (FAs) for many decades (McCulloch and Pitts, 1943; Kleene, 1956; Cleeremans et al., 1989; Siegelmann, 1996; Weiss et al., 2019; Korsky, 2019). Many works explore the extraction of FAs from trained RNNs (Giles et al., 1992; Das and Mozer, 1993; Kolen, 1993; Omlin and Giles, 1996; Giles et al., 1999; Weiss et al., 2018a) (note that our work hints at the possibility to extract FAs also from LTs). Others use synthetic and formal languages

to benchmark RNNs (Allen, 1990; Schmidhuber et al., 1999). The connection between RNNs and FAs also motivates certain architectural enhancements of RNNs, such as stack-augmented RNNs (Pollack, 1990; Das et al., 1992; Sun et al., 1993; Joulin and Mikolov, 2015; Grefenstette et al., 2015; DuSell and Chiang, 2020).

For further references on theoretical works studying RNNs, see also Merrill (2019, 2020); Ackerman and Cybenko (2020), and for Transformers, see also Hahn (2020); Weiss et al. (2021); Liu et al. (2022); Yun et al. (2020); Pérez et al. (2021).

## B   Review of Linear Attention vs. FWPs

Here we briefly review the formal connection between the "RNN-form" of LTs shown in Sec. 2 and attention (Katharopoulos et al., 2020; Schlag et al., 2021; Ba et al., 2016). Starting from Eq. 5, using the definition of $\boldsymbol{W}_t$ (Eq. 2) and $\boldsymbol{z}_t$ (Eq. 4),

$$\boldsymbol{y}_t = \frac{1}{\boldsymbol{z}_t \cdot \phi(\boldsymbol{q}_t)} \boldsymbol{W}_t \phi(\boldsymbol{q}_t) \tag{13}$$

$$= \frac{\left( \sum_{\tau=1}^{t} \boldsymbol{v}_\tau \otimes \phi(\boldsymbol{k}_\tau) \right) \phi(\boldsymbol{q}_t)}{\left( \sum_{\tau'=1}^{t} \phi(\boldsymbol{k}_{\tau'}) \right)^\mathsf{T} \phi(\boldsymbol{q}_t)} \tag{14}$$

$$= \frac{\sum_{\tau=1}^{t} \boldsymbol{v}_\tau \phi(\boldsymbol{k}_\tau)^\mathsf{T} \phi(\boldsymbol{q}_t)}{\sum_{\tau'=1}^{t} \phi(\boldsymbol{k}_{\tau'})^\mathsf{T} \phi(\boldsymbol{q}_t)} \tag{15}$$

$$= \sum_{\tau=1}^{t} \alpha_{t,\tau} \boldsymbol{v}_\tau \tag{16}$$

where $\alpha_{t,\tau} = \frac{\phi(\boldsymbol{k}_\tau)^\mathsf{T} \phi(\boldsymbol{q}_t)}{\sum_{\tau'=1}^{t} \phi(\boldsymbol{k}_{\tau'})^\mathsf{T} \phi(\boldsymbol{q}_t)}$. We can recognise that this is effectively attention with *normalised* weights $\alpha_{t,\tau}$ using $\phi(\boldsymbol{k}_t), \boldsymbol{v}_t, \phi(\boldsymbol{q}_t)$ as key, value, query vectors. For LTs/FWPs without normalisation of Eq. 5 (i.e., Eqs. 1-3), the derivation above is similar, but the corresponding attention weights are not normalised.

Note that this relation is analogous to the one that connects the perceptron to kernel machines (Aizerman et al., 1964; Irie et al., 2022a).

## C   Further details

### C.1   Further Comments on Proofs

Here we provide some more comments on the proofs of our propositions presented in Sec. 3.

For both Proposition 3.1 and Proposition 3.2, the original proofs, i.e., the proof of Merrill et al. (2020)'s Theorem 15 and that of Bhattamishra et al. (2020a)'s Proposition 4.1 respectively, consist in constructing a self-attention layer capable of

solving certain counting tasks—a task checking whether two alphabets appear the same number of times in a sequence, in the former, and the Shuffle-$k$ languages in the latter. In both cases, as softmax is not explicitly required for computing normalised attention weights, the proofs directly remain valid for NLTs. Now, the question is whether they still hold for ULTs: is the normalisation of attention weights required? The answer is *no* in both constructions. The core function of the layer consists in counting and comparing the occurrence of two symbols (e.g., opening and closing brackets in the case of Shuffle-$k$). The actual comparison is done by computing the difference between the two counts and comparing it against zero. This function is preserved without normalisation of attention weights. Therefore, these proofs can be directly adopted for ULTs.

For Proposition 3.3, the original proof of Bhattamishra et al. (2020a)'s Lemma C.4 shows that Transformers without positional encoding can not recognise $(aa)^*$ because the output of the Transformer is the same for all steps for this language defined using a single symbol. There is no way to distinguish between odd and even steps, which is essential to recognise $(aa)^*$. While this remains true for normalised linear attention resulting in uniform attention weights, this argument does not directly hold for unnormalised variants. Nevertheless, if we assume an extra layer normalisation layer following the self-attention layer (which is typically the case in practice), the constant-output argument also holds for unnormalised linear attention.

Finally, for Proposition 3.4, Merrill et al. (2020)'s proof for their Theorem 16 consists in "counting" the number of configurations of layer activations. This is independent of normalisation schemes, and remains valid for both unnormalised and normalised linear Transformers.

### C.2   Experimental Details

**Task Definition.**   For parity, $(aa)^*$, and $(abab)^*$, the output to be predicted at each step is the result for the prefix presented so far. For example for parity, if the input sequence is `0010`, the output sequence should be `TTFF`, where `T` and `F` denote 'true' and 'false' for parity. For Shuffle-2 (Suzgun et al. (2019a): a mixture of two Dyck-1 languages, i.e., with two kinds of parentheses), we encode the task as follows. By denoting the parenthesis () as type-0 and [] as type-1, we consider four cases: '0'

Table 3: Hyper-parameter search space.

| Parameters | Values |
|---|---|
| Number of layers | {1, 2, 4} |
| Hidden size | {8, 16, 32} |
| Feedforward block multiplier | {1, 2, 4} |
| Number of heads | {1, 2, 4} |
| Learning rate | {1e-2, 2e-2, 3e-2, 1e-3, 2e-3, 3e-3} |
| Batch size | {16, 32, 64} |

(both are closed), '1' (type-0 is open), '2' (type-1 is open), '3' (both are open); for example, for the input ([]), the output should be 2320.

**Dataset.** We use the official (pre-generated) dataset made publicly available by Bhattamishra et al. (2020a), except for reset Dyck-1 which we generate ourselves using their official public code. "Bin0" split contains sequences of "lengths" shorter than 50, while "Bin1" contains those with "lengths" between 51 and 100. The exact definition of "length" above depends on the tasks; for tasks such as parity, it directly refers to the actual sequence length; for tasks such as $a^n b^n c^n$, it refers to $n$, i.e., the actual lengths of Bin0 sequences are up to 150 for $a^n b^n c^n$, while they are between 153 and 300 for Bin1 sequences.

**Hyper-parameter search** spaces for all Transformer family models (i.e., all models except LSTM and e-LSTM) are shown in Table 3. Note that "Feedforward block multiplier" refers to the factor $N_{\text{ff}}$ that relates the hidden size $d_{\text{model}}$ of the Transformer to its feedforward up-projection size $d_{\text{ff}}$, i.e., $d_{\text{ff}} = N_{\text{ff}} d_{\text{model}}$. For LSTM and e-LSTM, we use the same search space except that the number of layers is in {1, 2}, and the hidden size is in {8, 16, 32, 64}, and irrelevant parameters (i.e., the feedforward block multiplier and the number of heads) are ignored. The reported results are the best performance across all the hyper-parameter search, as done in previous work (Bhattamishra et al., 2020a). Tables 4 and 5 display the best hyper-parameter configurations on each task for Recurrent Delta and SRWM models, respectively. For further details, we refer to our public code.

Any other configurations for the SRWM follow those of Irie et al. (2022c), except that we initialise the 'query' projection sub-matrix in the self-referential weight matrix using a normal distribution with a mean value of 0 and a standard

deviation of $0.01/\sqrt{d_{\text{head}}}$ while other sub-matrices use an std of $1/\sqrt{d_{\text{head}}}$ (this is motivated by the fact that a generated query vector is immediately multiplied with the same SRWM to produce a value vector).

### C.3 Details of e-LSTM

In the main text, we evaluate the element-wise LSTM with tied input-forget gates (e-LSTM; Irie et al. (2023)) as an illustrative example of computationally limited RNNs. e-LSTM is essentially an LSTM with only element-wise recurrence, which can be seen as a Quasi-RNN (Bradbury et al., 2017) with element-wise recurrent gate functions. Here we provide its detailed description. Let $d_{\text{in}}$ and $d_{\text{out}}$ denote positive integers. At each time step $t$, e-LSTM transforms an input vector $\boldsymbol{x}(t) \in \mathbb{R}^{d_{\text{in}}}$ to a recurrent hidden state $\boldsymbol{c}(t) \in \mathbb{R}^{d_{\text{out}}}$ as follows:

$$\boldsymbol{f}(t) = \sigma(\boldsymbol{F}\boldsymbol{x}(t) + \boldsymbol{w}^f \odot \boldsymbol{c}(t-1)) \qquad (17)$$

$$\boldsymbol{z}(t) = \tanh(\boldsymbol{Z}\boldsymbol{x}(t) + \boldsymbol{w}^z \odot \boldsymbol{c}(t-1)) \qquad (18)$$

$$\boldsymbol{c}(t) = \boldsymbol{f}(t) \odot \boldsymbol{c}(t-1) + (1 - \boldsymbol{f}(t)) \odot \boldsymbol{z}(t) \qquad (19)$$

$$\boldsymbol{o}(t) = \sigma(\boldsymbol{O}\boldsymbol{x}(t) + \boldsymbol{W}^o \boldsymbol{c}(t)) \qquad (20)$$

$$\boldsymbol{h}(t) = \boldsymbol{o}(t) \odot \boldsymbol{c}(t) \qquad (21)$$

where $\boldsymbol{f}(t) \in \mathbb{R}^{d_{\text{out}}}$, $\boldsymbol{z}(t) \in \mathbb{R}^{d_{\text{out}}}$, and $\boldsymbol{o}(t) \in \mathbb{R}^{d_{\text{out}}}$ are activations, $\boldsymbol{F} \in \mathbb{R}^{d_{\text{out}} \times d_{\text{in}}}$, $\boldsymbol{Z} \in \mathbb{R}^{d_{\text{out}} \times d_{\text{in}}}$, $\boldsymbol{O} \in \mathbb{R}^{d_{\text{out}} \times d_{\text{in}}}$ and $\boldsymbol{W}^o \in \mathbb{R}^{d_{\text{out}} \times d_{\text{out}}}$ are trainable weight matrices, and finally, $\boldsymbol{w}^f \in \mathbb{R}^{d_{\text{out}}}$ and $\boldsymbol{w}^z \in \mathbb{R}^{d_{\text{out}}}$ are trainable weight vectors.

Table 4: Best hyper-parameters for Recurrent Delta. When there are more than one best configurations, we report the one that converges the fastest.

| Parameters | Parity | $(aa)^*$ | $(abab)^*$ | $a^n b^n$ | $a^n b^n c^n$ | Shuffle-2 | Dyck-1 | Reset Dyck-1 |
|---|---|---|---|---|---|---|---|---|
| Number of layers | 1 | 1 | 2 | 1 | 1 | 4 | 1 | 1 |
| Hidden size | 4 | 8 | 8 | 8 | 16 | 16 | 16 | 8 |
| Feedforward block multiplier | 1 | 1 | 1 | 1 | 1 | 2 | 1 | 1 |
| Number of heads | 1 | 2 | 2 | 4 | 4 | 4 | 2 | 4 |
| Learning rate | 2e-2 | 2e-2 | 2e-2 | 3e-2 | 2e-2 | 2e-2 | 3e-2 | 3e-2 |
| Batch size | 16 | 16 | 16 | 16 | 16 | 32 | 16 | 16 |

Table 5: Best hyper-parameters for SRWM. When there are more than one best configurations, we report the one that converges the fastest.

| Parameters | Parity | $(aa)^*$ | $(abab)^*$ | $a^n b^n$ | $a^n b^n c^n$ | Shuffle-2 | Dyck-1 | Reset Dyck-1 |
|---|---|---|---|---|---|---|---|---|
| Number of layers | 1 | 2 | 1 | 2 | 1 | 1 | 1 | 1 |
| Hidden size | 8 | 16 | 16 | 16 | 8 | 8 | 16 | 16 |
| Feedforward block multiplier | 1 | 1 | 1 | 2 | 2 | 2 | 2 | 2 |
| Number of heads | 2 | 4 | 2 | 4 | 2 | 2 | 8 | 2 |
| Learning rate | 3e-2 | 2e-2 | 3e-2 | 1e-2 | 2e-2 | 3e-2 | 3e-2 | 3e-2 |
| Batch size | 16 | 16 | 16 | 16 | 16 | 32 | 16 | 16 |