# OpenReview forum: "Practical Computational Power of Linear Transformers and Their Recurrent and Self-Referential Extensions"
_EMNLP/2023/Conference — EMNLP 2023 Main_

### Official Review · Reviewer_Vzz8 · 2023-07-30

**Soundness:** 4

**Excitement:**

2: Mediocre: This paper makes marginal contributions (vs non-contemporaneous work), so I would rather not see it in the conference.

**Paper Topic And Main Contributions:**

This paper discusses the computational power of linear transformers (LTs) and fast weight programmers (FWPs). It also tests these methods and their extensions, such as delta-rule, recurrence, and self-reference, to see how effective they are in formal language recognition tasks.

**Questions For The Authors:**

It is generally a belief that LTs are weaker than vanilla Transformers. Can you make some discussions about that?

**Reasons To Accept:**

The paper theoretically reveals the computational power of LTs, and empirically verifies LTs' abilities on formal language recognition tasks. It is nice to have such results for reference.

**Reasons To Reject:**

Theoretical contributions is incremental since the proofs from previous studies remain valid for LTs. Additionally, experiments suggest that LTs inherit Transformer properties, which also appears trivial.

**Reproducibility:**

4: Could mostly reproduce the results, but there may be some variation because of sample variance or minor variations in their interpretation of the protocol or method.

**Reviewer Confidence:**

2: Willing to defend my evaluation, but it is fairly likely that I missed some details, didn't understand some central points, or can't be sure about the novelty of the work.

---

> ### Author Rebuttal · Authors · 2023-08-28
>
> We thank the reviewer for the valuable time spent on reviewing our work.
>
> > *"Theoretical contributions is incremental since the proofs from previous studies remain valid for LTs. "*
>
> Yes, our core technical contributions and “novelty” are mainly in the experimental section. At the same time, without highlighting these properties this way, we doubt that these results are trivial at all when the readers simply look at Eqs. 1-2;4-5 (we hope the reviewer concurs). Stating that these proofs remain valid (and also observing that they are no longer valid for the *unnormalised* case) is a necessary/important step. Overall, we think it is useful to have this “summary” of properties with specific pointers to the proof, which sets the stage to the main experimental section.
>
> > *"Additionally, experiments suggest that LTs inherit Transformer properties, which also appears trivial."*
>
> This statement is not fully accurate. It is true that the *vanilla* LTs directly inherits some of the standard Transformers’ properties discussed here, but this is not the case for their extensions.
> Our experiments show that the recurrent or self-referential extensions of linear Transformers enable them to recognise certain non-star-free languages including the parity. Since Transformers’ deficiency on the parity task is a very famous problem in this field (see Hahn 2020, inter alia), it is important to show that these architectural changes address this fundamental limitation. To the best of our knowledge, no prior work on linear Transformers has discussed these perspectives/benefits.
>
> > *"It is generally a belief that LTs are weaker than vanilla Transformers. Can you make some discussions about that?"*
>
> By *"weaker"* we assume that the reviewer is referring to the *empirical* performance
> (we have to be careful with the usage of the word *“weak”* here, since the paper is about computational capabilities).
>
> It is not a *“belief”*: there are many empirical evidences showing that the (in particular, vanilla) LTs underperform the standard Transformers (see, e.g. Schlag et al. 2021 which also show how the delta rule closes the gap between LTs and the standard Transformers).
> However, LTs have their unique benefits. In addition to their linear time-complexity advantage, LTs/FWPs have constant-size states which facilitate certain applications (e.g., reinforcement learning in game environments), and also bring novel/conceptual insights to Transformers as networks generating weights for other networks (for example, understanding in-context learning [von Oswald et al. Transformers learn in-context by gradient descent ICML 2023]). We generally consider the argument of type “this is believed to be weak, so it is not interesting” (we are not sure if this is what the reviewer is hinting at) is unfortunate as it may miss many opportunities and insights.
> If the reviewer wishes, we'll be happy to add a few sentences about this discussion in Appendix B.
>
> We hope that our response successfully draws the reviewer's attention to the core contributions of our empirical results (we respectfully point out that they are completely ignored in the original review), and improves the reviewer’s overall rating of this work.

---

### Official Review · Reviewer_AXki · 2023-08-08

**Soundness:** 2

**Excitement:**

2: Mediocre: This paper makes marginal contributions (vs non-contemporaneous work), so I would rather not see it in the conference.

**Paper Topic And Main Contributions:**

The paper investigates the computational expressivity of linear Transformers as well as several of its variants. By borrowing proofs from previous papers, the paper concludes that linear Transformers share some properties with the original Transformer. The paper conclude by performing experiments on linear Transformer's ability to model formal languages.

**Reasons To Accept:**

The paper considers a class of Transformers that so far has not been studied closely for its computational expressivity.

**Reasons To Reject:**

My main concern is the lack of novelty. All proofs in the paper are borrowed from previous works with one or two sentences of description. The authors should at least review the proof details in the Appendix so that the paper is at least self-contained. As it stands, it is difficult to judge the technical contribution as well as the correctness of the proofs in the paper.

**Reproducibility:**

3: Could reproduce the results with some difficulty. The settings of parameters are underspecified or subjectively determined; the training/evaluation data are not widely available.

**Reviewer Confidence:**

1: Not my area, or paper was hard for me to understand. My evaluation is just an educated guess.

---

> ### Author Rebuttal · Authors · 2023-08-28
>
> We thank the reviewer for the valuable time spent on reviewing our work.
> We acknowledge that this topic is not the reviewer’s area (“Reviewer Confidence: 1: Not my area”).
>
> > *"The authors should at least review the proof details in the Appendix so that the paper is at least self-contained."*
>
> This is indeed a good idea. This will also allow us to add more comments on WHY the proof in question does not hold for the unnormalized variants (we only hinted at this for one proposition in Line 239: *“For unnormalised LTs (Eqs. 1-3), certain proofs are not valid (e.g., the constant-output argument used for Prop. 3.3”*). We will add such a section in the appendix in the final version of this paper. Thank you for this suggestion.
>
> > *"My main concern is the lack of novelty. All proofs in the paper are borrowed from previous works with one or two sentences of description."*
> > *"As it stands, it is difficult to judge the technical contribution as well as the correctness of the proofs in the paper."*
>
> We would like to emphasise that our core technical contributions and “novelty” are mainly in the experimental part. Our experiments show that the recurrent or self-referential extensions of linear Transformers enable them to recognise certain non-star-free languages including the parity. Since Transformers’ deficiency on the parity task is a very famous problem in this field (see Hahn 2020, inter alia), it is important to demonstrate how these architectural changes address this fundamental limitation. To the best of our knowledge, no prior work on linear Transformers has discussed these perspectives/benefits.
>
> Also regarding the proofs, without highlighting these properties this way, we doubt that these results are trivial at all when the readers simply look at Eqs. 1-2;4-5 (we hope the reviewer concurs). Overall, we think it is useful to have this “summary” of properties with specific pointers to the proof, which sets the stage to the main experimental section.
>
> Hopefully, our response successfully directs the reviewer's attention to our experimental section and the corresponding contributions, which (as we respectfuly note) are *completely* ignored in the original review.

---

### Official Review · Reviewer_Dg9f · 2023-08-09

**Soundness:** 3

**Excitement:**

4: Strong: This paper deepens the understanding of some phenomenon or lowers the barriers to an existing research direction.

**Paper Topic And Main Contributions:**

The main contributions of the paper are: (1) This paper provides a theoretical analysis of the expressive power of normalised linear Transformers, showing that they inherit several capabilities and limitations of standard Transformers. For example, they can recognize certain counter languages, but not certain regular languages. (2) The paper empirically evaluates unnormalised linear Transformers and their extensions on several formal language recognition tasks, using both positive and negative examples. The paper demonstrates that recurrent and self-referential extensions can successfully solve and generalize on tasks that the standard linear Transformer fails on, such as parity, (aa) ∗ , and (abab) ∗ .

**Reasons To Accept:**

The paper provides both theoretical and empirical results that shed light on the properties and limitations of linear Transformers, a computationally efficient variant of standard Transformers.


**Reasons To Reject:**

(1) The paper focuses on a narrow set of formal language recognition tasks that may not reflect the real-world performance or challenges of linear Transformers or their extensions.
(2) The paper does not compare linear Transformers with other existing models that can also recognize formal languages, such as RNNs with external memory or stack-augmented RNNs.
(3) The paper does not provide enough details or analysis on how the recurrent and self-referential extensions affect the computational complexity or scalability of linear Transformers.

**Reproducibility:**

2: Would be hard pressed to reproduce the results. The contribution depends on data that are simply not available outside the author's institution or consortium; not enough details are provided.

**Reviewer Confidence:**

2: Willing to defend my evaluation, but it is fairly likely that I missed some details, didn't understand some central points, or can't be sure about the novelty of the work.

---

> ### Author Rebuttal · Authors · 2023-08-28
>
> We thank the reviewer for the valuable time spent on reviewing our work.
>
> > *"(1) The paper focuses on a narrow set of formal language recognition tasks that may not reflect the real-world performance or challenges of linear Transformers or their extensions."*
>
> This *“narrow”* set of formal language has been specifically selected based on the prior works exploring computational capabilities of Transformers (namely Bhattamishra et al. (2020)). This allowed us to focus on the most relevant formal languages.
>
> Regarding real-world performance vs. formal languages, this criticism generally applies to any works studying neural networks using formal languages (please see all the papers we cited in the introduction and Appendix A). Formal languages allow us to evaluate *in isolation* some *specific* computational abilities or shortcomings (such as counting abilities) that are the essence of more complex computations. These abilities/shortcomings are often difficult to identify when directly working on complex real-world tasks which involve a combination of various computational abilities. Beyond the scholastic values of answering the fundamental question of “what models can learn to compute”, experiments using formal languages allow us to analyse/characterise the impact of certain architectural changes on the model’s computational capabilities.
> In this work, we showed that the recurrent or self-referential extensions of linear Transformers fix the deficiency of the original models on non-star-free languages including the parity. Prior works focused on real-world applications have failed to identify these properties. Given that the Transformer’s deficiency on the parity task is a very famous problem in this field (see Hahn 2020, inter alia), it is important to demonstrate that these architectural changes address this fundamental limitation.
>
> > *"(2) The paper does not compare linear Transformers with other existing models that can also recognize formal languages, such as RNNs with external memory or stack-augmented RNNs. "*
>
> Here our focus is on tasks/formal languages on which the vanilla LTs and Transformers fail (based on Bhattamishra et al. 2020's prior work), to demonstrate how the three extensions enhance the LTs' capabilities. As can be seen in Table 1, the vanilla LSTM can perfectly recognise all these tasks, and any more powerful models (not shown here) can also do so. So further comparison with memory-augmented RNNs will not add much more value into this picture (than comparing to the LSTM baseline) within the scope of this work.
>
> Nethertheless, going beyond the scope of this work, further (theoretical) investigations into recurrent or self-referential extensions may benefit from proofs (e.g., by Merrill et al. 2020) or tasks (e.g. Delétang et al (2023); see also our limitation section Line 351) used in the existing works studying computational abilities of memory-augmented RNNs. We leave this for future work. We will be happy to mention this more explicitly in our limitation section (starting from Line 340).
>
> > *"(3) The paper does not provide enough details or analysis on how the recurrent and self-referential extensions affect the computational complexity or scalability of linear Transformers."*
>
> We’d like to emphasise that the architectures themselves are not a contribution of this work. They have been proposed in prior works (Schlag et al, 2021, Irie et al, 2021/2022c) *along with discussions on their computational complexity*. In fact, the result is simple: *all* these models have a linear time complexity w.r.t. the sequence length (this is why they are “linear” Transformers) unlike the standard Transformers with the quadratic complexity. Naturally, actual per-time-step computational costs differ from one model to another (which also depend on the actual implementation). For further discussions on wall-clock time comparisons etc., we directly refer to Schlag et al. (2021) and Irie et al. (2021) which have dedicated discussions about this.
> Here we focus on “computational abilities” of these models which have never been discussed in prior works.
>
> Finally, regarding the “Reproducibility” score of 2, as we promised in the first page of our paper (footnote 1), we will release our entire code publicly if the paper is accepted. As we specified in Line 704, the used dataset is also from the public repository by Bhattamishra et al. (2020); we will specify the exact link in our public repository. Hopefully this will improve the reviewer’s view on the reproducibility of this work.
>
> We hope our response brings clarifications to the major concerns raised by the reviewer.
> If you find our response useful, we will appreciate it a lot if the reviewer can consider increasing the scores. Thank you.

---

### Official Review · Reviewer_5khN · 2023-08-09

**Soundness:** 3

**Excitement:**

3: Ambivalent: It has merits (e.g., it reports state-of-the-art results, the idea is nice), but there are key weaknesses (e.g., it describes incremental work), and it can significantly benefit from another round of revision. However, I won't object to accepting it if my co-reviewers champion it.

**Paper Topic And Main Contributions:**

In this paper, the authors discuss the computational power of Transformers with linearised attention (Linear Transformers) in the context of formal languages.
The authors show that Linear Transformers inherit several limitations and capacities of the vanilla transformer: failing on non-star regular languages and performing well on counter languages.
The authors also show that the delta-rule does not improve these results but recurrence and self-reference do.

**Reasons To Accept:**

- The paper is clear and well written.
- The paper provides some interesting results.

**Reasons To Reject:**

- I do not see how these observations can lead to advances in the field.

**Reproducibility:**

4: Could mostly reproduce the results, but there may be some variation because of sample variance or minor variations in their interpretation of the protocol or method.

**Reviewer Confidence:**

1: Not my area, or paper was hard for me to understand. My evaluation is just an educated guess.

---

> ### Author Rebuttal · Authors · 2023-08-28
>
> We thank the reviewer for the valuable time spent on reviewing our work.
>
> > *"I do not see how these observations can lead to advances in the field."*
>
> We acknowledge that this research topic is not the reviewer’s area (“Reviewer Confidence: 1: Not my area”), and therefore, we can understand that this type of question may arise (even though, generally speaking, we respectfully note that some might perceive such a statement as highly offensive).
>
> We first would like to point out that the research approach studying neural networks using formal languages is a well-established domain. Our work builds upon many prior works (please refer to our introduction as well as Appendix A for further references). It consists of studying capabilities of neural architectures to learn to recognise formal languages.
>
> Formal languages allow us to evaluate *in isolation* some *specific* computational/generalisation abilities or shortcomings (such as counting abilities) that are the essence of more complex computations. These abilities/shortcomings are often difficult to identify when directly working on complex real-world tasks which involve a combination of various computational abilities. Beyond the scholastic values of answering the fundamental question of “what models can learn to compute”, experiments using formal languages allow us to analyse/characterise the impact of certain architectural changes on the model’s computational capabilities.
> For example, in this work, we showed that the recurrent or self-referential extensions of linear Transformers fix the deficiency of the original models on non-star-free languages including the parity. Prior works focused only on real-world applications have failed to identify these properties. Given that the Transformer’s deficiency on the parity task is a very famous problem in this field (see Hahn 2020, inter alia), it is important to demonstrate that these architectural changes address this fundamental limitation.
>
> We hope that our response brings novel perspectives to the reviewer's vision and perception of research on neural networks using formal languages. This topic, as evidenced by the numerous prior works we have cited, holds significant interest within the EMNLP/*ACL community.

---

### Official Review · Reviewer_RMs4 · 2023-08-10

**Soundness:** 3

**Excitement:**

4: Strong: This paper deepens the understanding of some phenomenon or lowers the barriers to an existing research direction.

**Paper Topic And Main Contributions:**

In their paper, the authors study variants of auto-regressive Transformers with linearized attention, focusing on linear Transformers (LT).
They argued that previous work on the language capabilities (recognition of formal languages) of RNN and Transformer architectures also holds for normalized LTs (Section 3).
Further, they empirically study the capabilities of unnormalized LTs using the challenge sets studied by Bhattamishra et al. (2020) and further examine three recently published architecture extensions in their experiments (delta-rule, recurrence, self-reference).
They show improvements in non-star-free regular language examples by recurrence and self-reference.
Further, they show that the delta rule allows LTs to recognize Reset Dyck-1, which is not possible without this extension.


**Questions For The Authors:**

(A) The task description in 4.1 defines an output encoding different from the work cited before. Is there a reason to not use the same prediction scheme, sigmoid over possible classes and enumerating all classes that are possible (having a value greater than 0.5), instead of introducing class N for non-predictable?


**Reasons To Accept:**

Interesting work that combines theoretical aspects and empirical experiments for a better understanding of the capabilities of linear Transformers learning to recognize formal languages.
With their experiments, they give evidence that the mentioned extensions improve performance on some tasks and lead to higher capabilities of the LT architecture.


**Reasons To Reject:**

The authors state in the beginning that many well-known results transfer directly from Transformers to linear Transformers.
The four properties were rather briefly listed with notes on "trivial" proofs that follow previous work.

The results in the paper are described as accuracies of various models but solely from this description, it remains unclear to me, whether these results are single-run experiments or whether they refer to best-k results from all corresponding configurations via hyperparameter search, as done in previous work.
Accordingly, it is difficult to decide how much significance the results finally have.


**Reproducibility:**

4: Could mostly reproduce the results, but there may be some variation because of sample variance or minor variations in their interpretation of the protocol or method.

**Reviewer Confidence:**

2: Willing to defend my evaluation, but it is fairly likely that I missed some details, didn't understand some central points, or can't be sure about the novelty of the work.

**Typos Grammar Style And Presentation Improvements:**

In section 4.1 task, prior work refers to Gers et al. 2000. I wonder, whether Gers et al. 2001 would fit better at this position. Also using a different encoding seems rather confusing, without giving an argument here.

The motivation for two different tables should be more clear: why do they have different structures and why are experiments split into main vs Reset Dyck-1?

L256: what CERTAIN languages? --> would be better to specify the described languages more concretely
L300: solution_s
LL692: using input samples that correspond to the defined alphabet improves the comprehensibility, rather "aabb" than "0011"
L250 & Table1: should refer to non-star-free regular languages instead

---

> ### Author Rebuttal · Authors · 2023-08-28
>
> We thank the reviewer for the valuable time spent on reviewing our work and for very relevant/useful comments.
>
> > “*The four properties were rather briefly listed with notes on "trivial" proofs that follow previous work.*”
>
> Yes, the reviewer is right.
> At the same time, without highlighting these properties this way, we doubt that these results are trivial at all when the readers simply look at Eqs. 1-2;4-5.
> Overall, we consider that it is necessary (even unavoidable) and useful to have this “summary” of properties with specific pointers to the proof, which sets the stage to the main experimental section.
>
> Following Reviewer AXki’s suggestion, we plan to add a section in the appendix that highlights the key idea of each proof. This would also allow us to more specifically explain why the proof does not hold for the *unnormalized* variant (which we only briefly mention for Prop. 3.3 Line 239 *“For unnormalised LTs (Eqs. 1-3), certain proofs are not valid (e.g., the constant-output argument used for Prop. 3.3”*)). Hopefully this will also alleviate the reviewer’s concern (if the reviewer has any other suggestions, we’ll be very happy to take).
>
> > *”it remains unclear to me, whether these results are single-run experiments or whether they refer to best-k results from all corresponding configurations via hyperparameter search, as done in previous work.”*
>
> Thank you for asking this clarification question: the reported results are the best performance across all the hyper-parameter search, as done in previous work.
> While we specified our hyper-parameter search space (in Appendix C, Line 717), the reviewer is right to point out that this specific information has not been explicitly mentioned. We’ll add this in the final version. In fact, we should also specify the best configuration found for each case. We will add a big table displaying the best hyper-parameter configuration for each model for each task in the appendix. Thank you very much for drawing our attention to this important aspect.
>
> > *”In section 4.1 task, prior work refers to Gers et al. 2000. I wonder, whether Gers et al. 2001 would fit better at this position.”*
>
> Thank you very much for pointing this out. This is effectively a typo. We will replace “Gers et al. 2000” by “Gers et al. 2001” in the final version.
>
> > *”(A) The task description in 4.1 defines an output encoding different from the work cited before. Is there a reason to not use the same prediction scheme, sigmoid over possible classes and enumerating all classes that are possible (having a value greater than 0.5), instead of introducing class N for non-predictable?”*
>
> Thank you for this expert question. It should be noted that the reference in question is rather old (Gers et al. 2001). In the modern context, we find it more natural to define the task as a simple *one-to-one sequence mapping problem*. This unifies the formulation across all the tasks used in this work, also allowing us to use the same code for all the tasks. Since this preserves the fundamental nature of the task, we opted for this formulation which we find more natural/convenient/elegant.
>
> > *”The motivation for two different tables should be more clear: why do they have different structures and why are experiments split into main vs Reset Dyck-1?”*
>
> Unlike in the experiments of Table 1, the hyper-parameter search space (specified in Appendix C) for the Reset Dyck-1 experiments (Table 2) is restricted to the *single-layer* case (following Bhattamishra al. (2020)’s observation that single-layer transformers fail at this task). This difference explains the separation into two tables. In the end, this also turned out to be good for space management, as Table 1 is already very large.
>
> Regarding the structure, the main goal of this table is to highlight the benefit of the delta rule; Table 2 compactly shows this in isolation (any other models are irrelevant for this discussion). Nevertheless, if the reviewer still thinks it's better to use exactly the same structure in Table 2 as in Table 1, we’ll be happy to change that in the final version.
>
> > *”L256: what CERTAIN languages? --> would be better to specify the described languages more concretely”*
>
> We agree with the reviewer. We’ll replace it by an explicit sentence: “for $a^nb^n$ and $a^nb^nc^n$, we define…”. Thank you for pointing this out.
>
> > *”L300: solution\_s”*
>
> Thank you for pointing this out. We’ll fix it.
>
> > *’LL692: using input samples that correspond to the defined alphabet improves the comprehensibility, rather "aabb" than "0011"’*
>
> Since this example is for the parity task and using alphabets used in other tasks (“a” and “b”) may give rise to other confusions, we’ll use “T” and “F” as in True and False.
> That is, Line 694 becomes “For example for parity, if the input sequence is 0010, the output sequence should be TTFF where “T” and “F” denote “true” and “false”’ for parity.” Thank you for pointing this out.
>
> > *”L250 \& Table1: should refer to non-star-free regular languages instead”*
>
> Yes, absolutely. We will fix it. Thank you very much.
>
> We hope our response above brings clarifications to all the concerns raised by the reviewer.
> If you find our response useful, we will appreciate it a lot if the reviewer can consider increasing the scores. Thank you.

---

### Meta-Review · Area_Chair_9URM · 2023-09-15

**Recommendation:** 4

**Metareview:**

**Summary of Reviews and Post-Rebuttal Discussion:**

Reviewer RMs4 appreciated the combination of theoretical and empirical aspects of the paper and acknowledged the deeper understanding it offers on the capabilities of linear Transformers. However, they raised concerns regarding the clarity of certain results, whether results were from single-run experiments or hyperparameter searches, and pointed out specific issues related to citations, task descriptions, and the overall presentation. The authors addressed these concerns in their rebuttal, providing explanations for their design choices, acknowledging and promising to fix some issues, and clarifying certain aspects of their experimental design.

Reviewer 5khN recognizes the clarity and well-written nature of the paper, and commends it for its interesting results. The main criticism stems from uncertainty about how the findings might advance the field, which the authors addressed in their rebuttal by pointing out the importance and relevance of studying neural networks using formal languages.

Reviewer Dg9f praised the paper for its theoretical analysis of normalized linear Transformers' expressive power and their empirical evaluations on formal language recognition tasks. The paper effectively provides insights into the properties and limitations of linear Transformers, a computationally efficient alternative to standard Transformers. Criticisms include the paper's limited focus on certain formal language tasks, not comparing linear Transformers to models like RNNs with external memory, and not thoroughly exploring the computational complexity of certain extensions. In response, the authors argued that their chosen languages are based on prior research and are meant to assess specific computational capabilities. They defended the absence of comparisons by focusing on tasks where traditional linear Transformers falter and clarified that their main contribution isn't the architecture but the insights provided, with computational complexities discussed in earlier research.

Reviewer Vzz8 highlighted that while the theoretical aspects are incremental, the paper makes novel contributions in the experimental section. The experiments indicate that the extensions of linear Transformers allow the model to recognize specific non-star-free languages. The authors defended their contributions in the rebuttal, emphasizing the importance of their results.

Reviewer AXki criticized the submission for its lack of novelty, noting an over-reliance on proofs from previous works. The authors defended their work, emphasizing the technical contributions and novel findings in the experimental section, particularly about the capabilities of linear Transformers to recognize non-star-free languages. Despite the authors' rebuttal, the reviewer remained skeptical about the paper's novelty, citing unfamiliarity with the extensively cited prior works.

It should be noted that overall confidence of reviewers was very low. Reviewers 5khN and AXki reported confidence 1 and the rest reported confidence 2. Still, reviewers mainly align in their judgment of soundness and excitement of the paper.

**Soundness:**
The paper appears to be mostly sound in its claims and arguments. The reviewers' concerns were mainly about clarity and presentation rather than the validity of the results. The authors have proposed solutions to address the clarity issues.

**Excitement:**
There is a mixed reception to the paper's excitement level. Some reviewers have reservations about the paper's potential to advance the field. However, the authors' rebuttal was strong, indicating the importance of studying neural architectures with formal languages.

---

### Decision · Program_Chairs · 2023-10-07

**Decision:**

Accept-Main

**Comment:**

**Summary of Reviews and Post-Rebuttal Discussion:**

Reviewer RMs4 appreciated the combination of theoretical and empirical aspects of the paper and acknowledged the deeper understanding it offers on the capabilities of linear Transformers. However, they raised concerns regarding the clarity of certain results, whether results were from single-run experiments or hyperparameter searches, and pointed out specific issues related to citations, task descriptions, and the overall presentation. The authors addressed these concerns in their rebuttal, providing explanations for their design choices, acknowledging and promising to fix some issues, and clarifying certain aspects of their experimental design.

Reviewer 5khN recognizes the clarity and well-written nature of the paper, and commends it for its interesting results. The main criticism stems from uncertainty about how the findings might advance the field, which the authors addressed in their rebuttal by pointing out the importance and relevance of studying neural networks using formal languages.

Reviewer Dg9f praised the paper for its theoretical analysis of normalized linear Transformers' expressive power and their empirical evaluations on formal language recognition tasks. The paper effectively provides insights into the properties and limitations of linear Transformers, a computationally efficient alternative to standard Transformers. Criticisms include the paper's limited focus on certain formal language tasks, not comparing linear Transformers to models like RNNs with external memory, and not thoroughly exploring the computational complexity of certain extensions. In response, the authors argued that their chosen languages are based on prior research and are meant to assess specific computational capabilities. They defended the absence of comparisons by focusing on tasks where traditional linear Transformers falter and clarified that their main contribution isn't the architecture but the insights provided, with computational complexities discussed in earlier research.

Reviewer Vzz8 highlighted that while the theoretical aspects are incremental, the paper makes novel contributions in the experimental section. The experiments indicate that the extensions of linear Transformers allow the model to recognize specific non-star-free languages. The authors defended their contributions in the rebuttal, emphasizing the importance of their results.

Reviewer AXki criticized the submission for its lack of novelty, noting an over-reliance on proofs from previous works. The authors defended their work, emphasizing the technical contributions and novel findings in the experimental section, particularly about the capabilities of linear Transformers to recognize non-star-free languages. Despite the authors' rebuttal, the reviewer remained skeptical about the paper's novelty, citing unfamiliarity with the extensively cited prior works.

It should be noted that overall confidence of reviewers was very low. Reviewers 5khN and AXki reported confidence 1 and the rest reported confidence 2. Still, reviewers mainly align in their judgment of soundness and excitement of the paper.

**Soundness:**
The paper appears to be mostly sound in its claims and arguments. The reviewers' concerns were mainly about clarity and presentation rather than the validity of the results. The authors have proposed solutions to address the clarity issues.

**Excitement:**
There is a mixed reception to the paper's excitement level. Some reviewers have reservations about the paper's potential to advance the field. However, the authors' rebuttal was strong, indicating the importance of studying neural architectures with formal languages.